# Effect of a 16-Session Qigong Program in Non-Hodgkin Lymphoma Survivors: A Randomized Clinical Trial

**DOI:** 10.3390/jcm11123421

**Published:** 2022-06-14

**Authors:** Keyla Vargas-Román, Emilia I. De la Fuente-Solana, Jonathan Cortés-Martín, Juan Carlos Sánchez-García, Christian J. González-Vargas, Lourdes Díaz-Rodríguez

**Affiliations:** 1Research Group CTS1068, Andalusia Research Plan, Junta de Andalucía, 18014 Granada, Spain; jonathan.cortes.martin@gmail.com (J.C.-M.); jsangar@ugr.es (J.C.S.-G.); cldiaz@ugr.es (L.D.-R.); 2Spanish Education Ministry Program FPU16/01437, Methodology of Behavioral Sciences Department, Faculty of Psychology, University of Granada, 18071 Granada, Spain; 3Methodology of Behavioral Sciences Department, Faculty of Psychology, University of Granada, 18071 Granada, Spain; edfuente@ugr.es; 4Department of Nursing, Faculty of Health Sciences, University of Granada, 18016 Granada, Spain; 5Fulton Schools of Engineering Department, Faculty of Computing and Augmented Intelligence, Arizona State University, Tempe, AZ 85281, USA; cjgonz21@asu.edu

**Keywords:** lymphoma cancer, HRV, autonomic nervous system, non-Hodgkin lymphoma, anxiety, depression

## Abstract

Background: The treatment associated with non-Hodgkin lymphoma patients may cause adverse effects on their physical and psychological condition. The aim of this study is to detect the response to an eight-week, 16-session, 60-min presential Qigong program in anxiety, depression and vagal nerve activity alongside a control group. Methods: A randomized controlled clinical trial was managed. Randomization was carried out by generating a numerical sequence of three cycles through the software EPIDAT 4.1. Numbers were placed in sealed opaque envelopes for assignment to the different groups. Results: Anxiety levels were substantially decreased in the experimental group, with a large effect size (F = 30.38, *p* < 0.00). Depression levels had an improvement in the experimental group in contrast to the control group, reaching statistical significance (F = 19.19, *p* < 0.00). Heart Rate Variability unveiled significant results in terms of between-group differences, with a large effect size in the HRV Index (F = 15.80, *p* < 0.00), SDNN (F = 8.82, *p* < 0.00), and RMSSD (F = 6.72, *p* < 0.01) in the time domain, and a medium effect size in the HF (F = 9.78, *p* < 0.003), LF (F = 9.78, *p* < 0.00), and LF/HF Ratio (F = 18.44, *p* < 0.00) in the frequency domain, which were all bettered in the experimental group, after the Qigong program. Conclusions: Qigong therapy can be an effective therapeutic activity in consonance with traditional medicine to improve psychological health and autonomic nervous system balance in non-Hodgkin lymphoma survivors.

## 1. Introduction

Cancer is a broad group of illnesses that can start in practically any organ or tissue of the body when abnormal cells grow uncontrollably, breaching adjacent parts of the body or escalating to other organs [1]. Non-Hodgkin lymphoma had 544,352 new cases worldwide in 2020 in both genders according to the world health organization in the category of global hematic cancers. It ranks in the eleventh position among other types of cancers in 2020. In terms of its fatality rate, non-Hodgkin lymphoma cancer had 259,793 cases in 2020 in both genders worldwide [2].

In Spain, in 2020, non-Hodgkin lymphoma within blood cancers, was one of the highest diagnosed, being in the top nine positions. The estimated number of cases in Spain in 2020 was 66,733 [3]. Mortality from this tumor has reduced since the late 1990s, at a rate of 3% less mortality each year, showing a clear advancement in the efficiency of treatments [4].

The conventional medical treatments for non-Hodgkin lymphoma are chemotherapy and rafigurecell transplants [5]. These therapies may have high inimical consequences such as anxiety, depression, the loss of physical health, and a high risk of heart failure; these difficulties lead to a deteriorating quality of life [6]. Advances in the biomedical field have made improvements in the understanding of the origin of various cancers, along with treatments and prevention strategies [7]. Prevention is necessary in order to avoid possible risks in the development of cancer in general, including a good diet, exercise, and good sleep quality.

However, those patients who are already diagnosed need a treatment strategy to mitigate the symptoms of their cancer or its treatment. Symptoms may also include an imbalance in the autonomic nervous system, which is integrated by the sympathetic and parasympathetic nervous systems [8]. These are in addition to the vagus nerve responsible for the regulation of mood status, the immune system and heart rate [9]. Heart rate variability is the synergy between the autonomic nervous system and the cardiovascular system, which serves as a noninvasive biomarker of health [10]. Diminished heart rate variability (HRV) has been noted to be related to cardiac autonomic dysfunction [11]. Previous studies have shown an association between non-Hodgkin lymphoma and low HRV [12], which could be caused by the conventional medical treatment.

Complementary and integrative medicine treatments, including physical activity, yoga/Tai Chi, and meditation, in addition to traditional oncology treatment, may have a beneficial influence on psychological affliction, anxiety, pain, fatigue, and sleep disruption, leading to a bettered QoL in cancer patients [13]. In addition to the activities mentioned before, another mind–body exercise that is becoming popular is the Qigong. Qigong was originally developed in China, and is deeply rooted in traditional Chinese medicine. It involves a collection of precise and placid movements, and integrates the regulation of the breath and body control [14]. Studies have shown the benefits of practicing Qigong, improving fatigue, sleep quality, anxiety, depression and cardiotoxicity in cancer patients and non-Hodgkin lymphoma [14,15,16].

To our best knowledge, presently, there is no Qigong study that has targeted survivors with non-Hodgkin lymphoma for the betterment of patient heart rate variability (HRV), anxiety and depression outcomes. Therefore, the aim of the present study was to detect the effects of an eight-week presential Qigong program of 60 min on psychological parameters and vagal nerve activity with regard to non-Hodgkin lymphoma, and to compare the results with a control group that did not participate in the program.

## 2. Materials and Methods

### 2.1. Study Design

A randomized, single-blinded, controlled trial was conducted on non-Hodgkin lymphoma survivors. Randomization was carried out by generating a numerical sequence of 3 cycles through the software EPIDAT 4.1. The numbers were placed in sealed opaque envelopes for assignment to the different groups. Informed consent was collected from all of the candidates in the study, which was authorized by the local research ethics committee (CEI-GRC-9) and ensued the principles of the Declaration of Helsinki. The trial protocol is registered at ClinicalTrials.gov (NCT04701554).

### 2.2. Setting and Selection of the Participants

Patients from the Oncology Unit in the University Hospital Virgen de las Nieves in the province of Granada city were reached by the researchers face-to-face, by telephone, and by social media (Facebook, Instagram and Twitter). The inclusion criteria were a primary diagnosis of non-Hodgkin lymphoma cancer (Grades I–IIIa), being of any gender, being 18 years old or older, and having completed the primary section of traditional cancer treatment (chemotherapy, radiotherapy, surgery and/or immunotherapy) previously within 6 months to 1 year. The exclusion criteria for this study were the existence of metastasis and/or active cancer, a history of cardiovascular disease, and taking medications known to alter vagus nerve activity. Randomization was used to allocate the patients to the control or intervention groups. The flow of the participants through the study is shown in Figure 1. A single researcher (K.V.-R.) reached the participants by telephone, in order to collect epidemiological data related to their medical history and demographic characteristics, including their age, gender, race, social status, education level, occupation, alcohol and smoking customs, menopause status, type of cancer treatment, transplantation situation, cancer stage, weight, and height.

### 2.3. Control Group

The control group obtained advice on healthy habits, physical activity and dietary suggestions.

### 2.4. Experimental Group

The experimental group took part in the Qigong course over an interim of 2 months, for 16 sessions, with a total of 16 h.

### 2.5. Qigong Training Program

A meditation coach with more than a decade of knowledge ran a Qigong program at the Wudang Shan Center in Granada city. The non-Hodgkin survivors had an initial encounter with the instructor to talk about the Qigong program, emphasizing the concentration techniques, coordinated musculoskeletal movements, and diaphragmatic breathing. Over an eight-week period, two times a week, 60-min sessions—for a total of 16 h—were conducted each Tuesday and Friday in the afternoon. Ultimately, the objective was to learn how to focus attention on the breathing techniques and improving physical function.

Each 60-min session started with a confirmation sheet, and a brief consultation session of about 10 min to answer questions related to the Qigong practice. This was followed by 20 min of mild elongation and body motion in standing postures to trigger the body along the energy channels, and 10 min of motion in a seated posture (a Nei Yang Gong exercise for inner nurture, relaxing the head, neck, shoulders, waist, lower back, legs, and feet, and imagining an inner smile while in this posture). This concluded with 20 min of meditation, including breathing exercises, starting with abdominal breathing, chest breathing to bring energy regulation, relaxation, and the feeling of the Qi (vital energy). The researcher (K.V.-R.) reached the candidates weekly, by telephone, to send the participants reminders of their Qigong practices.

### 2.6. Sample Size Calculation

In order to estimate the sample size for statistical power, the EPIDAT 4.1 software (Xunta de Galicia, Santiago, Spain) was used. This was 80% with a 5% level of significance, based on formerly divulged data [14]. A margin sample size of 20 individuals per group was determined.

### 2.7. Outcome Measures

The data collection took place between week zero and the eighth week after the Qigong program. Both data collections were carried out by a single researcher (K.V.-R).

### 2.8. Hospital Anxiety and Depression Scale (HADS) Score

This authorized self-administered questionnaire is intended to detect the potential existence of anxiety and depression. It contains 14 items that equally measure anxiety and depression, with responses on a 4-point Likert scale (0–3); the responses are related to how the patient felt in the previous week [17].

### 2.9. Short-Term HRV

The participants first lay reclined in a quiet room (22–25 °C) for 10 min of cessation with normal breathing timed by a 0.2 Hz metronome. ECG signals were then acquired for 5 min using a Holter monitor with a fitted shunt channel II system (Norav Holter NR302, Braemar, Brunsville, MN, USA). Heart Rate Variability was computed from ECG recordings as the time interval between successive heart beats (RR interval). The following time-domain parameters were determined: the standard deviation of the mean normal-to-normal (NN) interval (SDNN), the square root mean square differences of successive NN intervals (RMSSD), and the number of all NN intervals divided by the maximum of all NN intervals (HRV index). The following spectral components were determined in the frequency domain: the low-frequency (LF) band (0.04–0.15 Hz), as a measure of sympathetic and parasympathetic activities; the high-frequency (HF) band (0.15 to 0.40 Hz), which is associated with vagal-parasympathetic activity; and the LF/HF ratio, which indicates sympathovagal balance. Spectral analysis was performed with the NH301-4 software (Norav, version 2.70), using fast Fourier transform algorithms. All of this procedure followed the recommendations of the Task Force of the European Society of Cardiology and the North American Society of Pacing and Electrophysiology, as published by Kuppussamy (2020) [18,19].

### 2.10. Statistical Analysis

For the statistical analysis, IBM-SPSS 26.0 was used. The results were defined as means with standard deviations for continuous variables, and percentages with 95% confidence intervals for categorical variables. *t*-tests and chi-square tests were performed for the continuous and categorical variables’ between-group differences at the baseline. The Shapiro–Wilk test was used to verify the data distribution normality. An ANCOVA test was performed to measure possible influences between covariates and baseline variables during the pre- and post-interventions, including the control and experimental groups.

## 3. Results

Initially, fifty patients enrolled in this study, and eleven survivors didn’t finalize the program. This left a final sample of 39 non-Hodgkin survivors, with 19 females and 20 males, with a mean (SD) age of 44.49 (10.60) years, a mean height of 170.21 (8.22) cm, a mean weight of 65.55 (9.06) kg, and therefore a mean BMI of 24.75 (4.21) kg/m^2^. All of the candidates except one were Caucasian, 51.3% were married, 48.7% had completed middle education, 43.6% were unemployed, 69.2% were non-smokers, 66.7% did not consume alcohol, 43.6% were in cancer stage II, 74.4% received chemotherapy, and only two patients had received transplants (Allotransplant). The only statistically significant difference in the above variables among the experimental (*n* = 20) and control (*n* = 19) groups was in the BMI, with a higher index in the control group versus the experimental group (Table 1).

In the experimental group, the anxiety levels were substantially decreased, with a large effect size (F = 30.38, *p* < 0.00) (Table 2). The depression levels had an improvement in the experimental group in contrast to the control group, reaching statistical significance (F = 19.19, *p* < 0.00). Heart Rate Variability unveiled significant results in terms of between-group differences, with a large effect size in the HRV Index (F = 15.80, *p* < 0.000), SDNN (F = 8.82, *p* < 0.00), and RMSSD (F = 6.72, *p* < 0.01), in the time domain, and a medium effect size in the HF (F = 9.78 *p* < 0.00), LF (F = 9.78 *p*< 0.00), and LF/HF Ratio (F = 18.44 *p* < 0.00) in the frequency domain, which were all bettered in the experimental group, after the Qigong program (Table 2). Covariates had no influence on these results.

## 4. Discussion

To our understanding, this is the first controlled trial to evidence the progression of the cardiovascular balance and psychological health of non-Hodgkin survivors after an eight-week Qigong program of 60 min per session twice a week (16 sessions in total), in contrast to a control group. Once the program ended, the non-Hodgkin lymphoma survivors demonstrated a decrease in anxiety and depression, and higher heart rate variability markers.

All of these compiled data add weight to past studies that demonstrated the effectiveness of complementary therapies by improving the heart rate variability in cancer survivors [16]. Regarding mental health, including anxiety and depression, other study emphasized the use of meditation therapies like mindfulness, which lowered these parameters in cancer patients [20]. It was previously found that an eight-week Qigong program reduced anxiety and depression in breast cancer survivors [15]. This confirmed that with traditional medical treatment and complementary therapies, patients can reduce their levels of anxiety and depression, and better their cardiovascular balance caused by the stress of the disease itself or the impact of medical treatments in the patients.

Moreover, past studies have shown that vagal activity in cancer patients tends to be impaired in comparison to healthy people [21]. Additionally, a previous case-controlled study described the cardiovascular imbalance in breast cancer survivors compared to gender-matched controls [22]. This difference might be explained by the emotional distress associated with the cancer [23]. In a recent published meta-analysis, it was shown that lymphoma patients are liable to suffer anxiety, mainly due to the remission stage of their disease or caused by the treatment itself [24]. This can explain the studies showcasing the fact that people are turning to non-pharmacological and non-conventional interventions [21] to battle the mental outcomes and physical problems that cancer diseases might cause.

Our results indicated that our non-Hodgkin survivors had reduced anxiety and depression levels after the Qigong sessions; this is in consonance with a previous study showing that an intervention with medical Qigong led to better mood status, specifically decreasing the anxiety and depression in cancer patients [25]. In the present research, the existence of anxiety correlated with a lower HRV value. Another significant finding is that a correlation between depression and HRV was present: the higher the depression, the lower the HRV values, confirming the relationship between good mental health and HRV. In the time domain, the SDNN, RMSSD, and HRV index values were substantially larger after the Qigong program, in comparison to controls who didn’t participated. Similar findings were reported in previous studies with cancer patients and a mindfulness therapy [20].

The positive effects of Qigong therapy regarding the improvement of mental health and HRV were previously described [14,15,16,26]. However, this is the first controlled clinical trial that demonstrated the improvement of psychological health and cardiovascular balance produced by a Qigong program in lymphoma patients, with a significant reduction in anxiety and depression, and an improvement in heart rate variability values.

Some study limitations that need to be included are, firstly, the sample size of the subjects, which reduces the extent to which the results can be extrapolated. The follow-up was quite short, and it is unknown whether the patients followed the therapeutic practice alone. The ethnicity was, in the majority, Caucasian in both groups; given the high diversity of ethnicities in Spain, the sample should have been more diverse. We should finalize with the stipulation that a larger sample size might confirm the improvement, such that the results can be extrapolated to a bigger population in future research.

## 5. Conclusions

An eight-week Qigong program of 60 min per session twice a week with a total of 16 sessions might be an effective way to improve the psychological health and cardiovascular balance in non-Hodgkin lymphoma survivors. This would work by reducing anxiety, depression, and heart rate variability parameters. Future clinical trials are assured to certify the effectiveness of a Qigong program to improve the quality of life of non-Hodgkin lymphoma survivors.

## Figures and Tables

**Figure 1 jcm-11-03421-f001:**
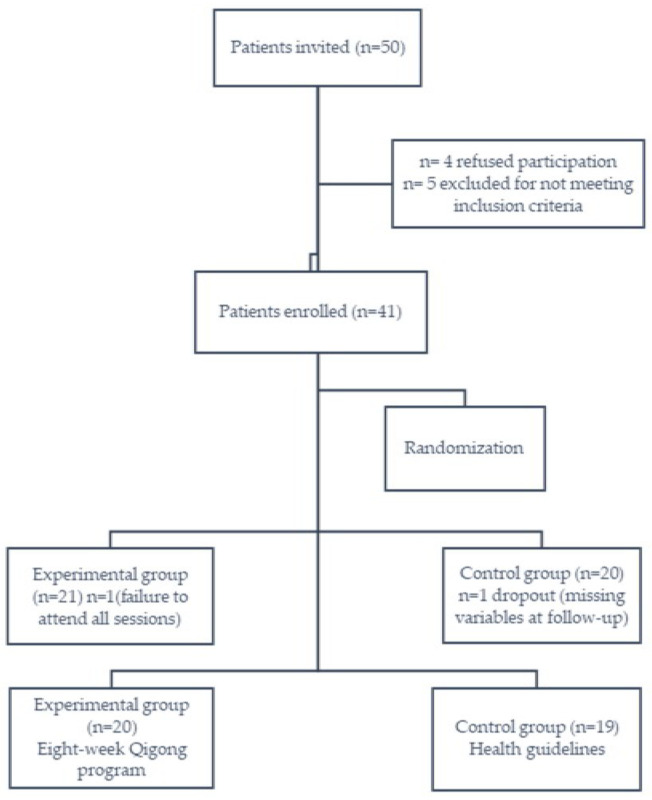
Flow of the participants.

**Table 1 jcm-11-03421-t001:** Sample characteristics and comparison between the study groups.

Variables	Control Group (*n* = 19)	Qigong Group (*n* = 20)	*p*
**Age (y) Mean (SD) ****	43.74 ± 10.53 (22–60)	45.20 ± 10.88 (23–68)	0.67
**Gender (%) ***			0.86
Female	47.4	50.0	
Male	52.6	50.0	
**Ethnicity (%) ***			0.32
Caucasian	100	95.0	
Other	0	5.0	
**Marital status (%) ***			0.83
Single	21.1	20.0	
Married	52.6	50.0	
Divorced	10.5	5.0	
Lives with partner	15.8	20.0	
Widowed	0	5.0	
**Educational level (%) ***			0.52
Primary studies	15.8	25.0	
Secondary studies	57.9	4.0	
Higher education	26.3	35.0	
**Occupational status (%) ***			0.20
Homemaker	10.5	0	
Employed	42.1	20.0	
Unemployed	31.6	55.0	
Retired	15.8	20.0	
Other	0	5.0	
**Smoking Status (%) ***			0.14
Smoker	0	10.0	
Non-smoker	63.2	75.0	
Ex-smoker	36.8	15.0	
**Alcohol status (%) ***			0.26
Don’t consume	57.9	75.0	
Consume monthly	31.6	25.0	
Consume weekly	10.5	0	
**Menopausal status (%) ***			0.93
NO	78.9	80.0	
YES	21.1	20.0	
**Type of treatment (%) ***			0.31
Chemotherapy	68.4	80.0	
Radiotherapy	10.5	15.0	
Immunotherapy	21.1	5.0	
**Cancer Stage (%) ***			0.74
I	15.8	20.0	
II	36.8	50.0	
III	15.8	10.0	
IV	31.6	20.0	
**Transplants (%) ***			0.97
Autologous transplant	0	0	
Allotransplant	5.3	5.0	
No transplant	94.7	95.0	
**Weight (kg) Mean (SD) ****	72.04 ± 8.94 (56–84)	67.18 ± 8.74 (45–81)	0.09
**Height (cm) Mean (SD) ****	172.47 ± 7.60 (159–186)	168.05 ± 8.40 (150–180)	0.09
**Body Mass Index Mean ** (Kg/m^2^) (SD)**	26.48 ± 4.23 (18–40)	23.10 ± 3.55 (17.8–32.4)	0.01 *

Values are expressed as means ± standard deviation (95% confidence interval). The Chi-square test * and Student *t*-test ** were used for between-group comparisons; * *p* < 0.05.

**Table 2 jcm-11-03421-t002:** Before- and after-treatment comparison between the outcomes.

Outcomes	Control Group (*n* = 19)	Qigong Group (*n* = 20)	F	*p*
**Hospital Anxiety Depression Scale values**				
Anxiety			30.38	0.00 *
Baseline	6.32 ± 4.30	8.35 ± 4.67		
Post-treatment	7.0 ± 4.35	4.00 ± 3.8		
Depression			19.19	0.00 *
Baseline	3.47 ± 2.59	4.55 ± 3.28		
Post-treatment	3.47 ± 2.93	2.65 ± 2.79		
**Heart Rate Variability**				
SDNN			8.82	0.00 *
Baseline	68.92 ± 49.21	61.86 ± 44.56		
Post-treatment	67.99 ± 66.54	123.40 ± 92.71		
RMSSD			6.72	0.01 *
Baseline	50.87 ± 34.62	72.60 ± 47.33		
Post-treatment	54.31 ± 35.81	140.53 ± 107.19		
HRV index			15.80	0.00 *
Baseline	11.67 ± 3.88	9.80 ± 4.89		
Post-treatment	10.14 ± 4.63	15.26 ± 4.98		
LF			11.77	0.00 *
Baseline	308.64 ± 313.33	1016.76 ± 1148		
Post-treatment	553.76 ± 399.26	405.29 ± 427.22		
HF			9.78	0.00 *
Baseline	497.38 ± 476.11	268.13 ± 345.10		
Post-treatment	380.44 ± 445.67	609.25 ± 448.42		
LF/HF RATIO			18.44	0.00 *
Baseline	1.48 ± 1.93	6.27 ± 6.67		
Post-treatment	5.25 ± 6.60	1.11 ± 1.38		

For comparisons between the interventions, * *p* < 0.05 SDNN = standard deviation of the normal-to-normal interval; RMSSD = root mean square of successive differences; LF = low frequency; HF = high frequency. ANCOVA = analysis of covariance was performed.

## Data Availability

Data is contained within the article.

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
