# Peer review of "Effect of a 16-Session Qigong Program in Non-Hodgkin Lymphoma Survivors: A Randomized Clinical Trial"

_jcm, 2022, doi:10.3390/jcm11123421_

Round 1

Reviewer 1 Report

This is a randomized, controlled clinical trial aimed at determining the effects of an eight-week, 60-minute, face-to-face Qigong program on psychological parameters and vagus nerve activity in non-Hodgkin's lymphoma, comparing the results with a control group that did not participate in the program. The method was adequate to reach the objectives and the steps were very detailed. The results will contribute to the advancement of knowledge in the field of oncology and benefit Non-Hodgkin Lymphoma survivors.

Suggestion: include more method data in the summary, how the sample was randomized and allocated, and data analysis

Author Response

Thank you for the comments regarding the paper, we took the comments into consideration and modified the manuscript accordingly. Hope this answers all the concerns presented. If any other concern is detected please don’t hesitate to ask.

Response to the review : 

Thank you for the suggestion, more details referring to the randomization and data analysis were added to the methods on the summary. 

Reviewer 2 Report

This is a randomized controlled clinical trial that followed the steps of conventional scientific methodology. The text follows the academic format and describes the ethical care of research. The conclusion contributes to the quality of life of survivors of an impacting disease. It also meets people's general yearning for non-pharmacological interventions.

The following demands are for the paper to conform to the standards of this journal. I ask that the authors please specify:

- the period in which the data collection took place (MM/YYYY - MM/YYYY); if it was during the covid-19 pandemic, one phrase about how social distancing and mortality rates were at that time would be useful (as these elements can interfere with mental balance).

- why the chosen control intervention was health education, as I imagine that a better intervention would be for patients to watch relaxing nature scenes in a group...

- in the study limitations paragraph (line 224), add that the follow-up was quite short and that it is unknown whether the patients followed the therapeutic practice alone (how about a new study on this, contacting the patients now?).

Author Response

Dear reviewer: 

Thank you for the comments regarding the paper, we took the comments into consideration and modified the manuscript accordingly. Hope this answers all the concerns presented. If any other concern is detected please don’t hesitate to ask.

Response: Thank you for the suggestions. Here are the specifications asked:

  • The period which the data collection took place was December 2019- May 2020. The results, weren’t affected since the qigong sessions were finished and the samples were taken right before the pandemic became serious.
  • The chosen control intervention was health education since, on the experimental group we had the education plus the qigong sessions. That is the usual procedure in Spain for patients diagnosed with cancer, a healthy diet, exercise and no stress.
  • We added the line that was suggested. A study following up these patients is planned in the near future.